# Anderson light localization in biological nanostructures of native silk

Seung Ho Choi [1], Seong-Wan Kim[2], Zahyun Ku[3], Michelle A. Visbal-Onufrak[1], Seong-Ryul Kim[2], Kwang-Ho Choi[2], Hakseok Ko[4,5], Wonshik Choi[4,5], Augustine M. Urbas[3], Tae-Won Goo[6] & Young L. Kim[1,7,8]

Light in biological media is known as freely diffusing because interference is negligible. Here, we show Anderson light localization in quasi-two-dimensional protein nanostructures produced by silkworms (*Bombyx mori*). For transmission channels in native silk, the light flux is governed by a few localized modes. Relative spatial fluctuations in transmission quantities are proximal to the Anderson regime. The sizes of passive cavities (smaller than a single fibre) and the statistics of modes (decomposed from excitation at the gain–loss equilibrium) differentiate silk from other diffusive structures sharing microscopic morphological similarity. Because the strong reflectivity from Anderson localization is combined with the high emissivity of the biomolecules in infra-red radiation, silk radiates heat more than it absorbs for passive cooling. This collective evidence explains how a silkworm designs a nanoarchitectured optical window of resonant tunnelling in the physically closed structures, while suppressing most of transmission in the visible spectrum and emitting thermal radiation.

[1] Weldon School of Biomedical Engineering, Purdue University, West Lafayette, IN 47907, USA. [2] Department of Agricultural Biology, National Institute of Agricultural Sciences, Rural Development Administration, Wanju 55365, Republic of Korea. [3] Materials and Manufacturing Directorate, Air Force Research Laboratory, Wright-Patterson Air Force Base, OH 45433, USA. [4] Center for Molecular Spectroscopy and Dynamics, Institute for Basic Science, Seoul 02841, Republic of Korea. [5] Department of Physics, Korea University, Seoul 02841, Republic of Korea. [6] Department of Biochemistry, School of Medicine, Dongguk University, Gyeongju 38066, Republic of Korea. [7] Regenstrief Center for Healthcare Engineering, Purdue University, West Lafayette, IN 47907, USA. [8] Purdue Quantum Center, Purdue University, West Lafayette, IN 47907, USA. Seung Ho Choi, Seong-Wan Kim and Zahyun Ku contributed equally to this work. Correspondence and requests for materials should be addressed to Y.L.K. (email: youngkim@purdue.edu)

When light waves undergo multiple scattering through inhomogeneous dielectric biomacromolecules, interference is conventionally ignored[1,2], because the phases of most scattered waves are uncorrelated and their interference is cancelled out. If the scattered waves yield an identical phase delay for destructive or constructive interference[3–5], the outgoing waves can be forbidden by off-resonance or occasionally being on-resonance with Anderson-localized modes[6,7]. Originally, Anderson localization has been used to describe the metal–insulator transition, resulting from scattering of the electronic wavefunction in random defects of the electronic potential[8]. Since this concept was proven for a system where the total energy is conserved, it has recently been extended to non-conservative bosonic fields (e.g. microwaves and light waves)[6,7,9–13]. For light waves, conclusive observation of three-dimensional (3D) Anderson localization (i.e. visible and infra-red ranges) still remains as an open problem[14]. Only when the system dimension is reduced, convincible demonstration of Anderson light localization has been reported[10,12]. No matter the dimensionality, this phenomenon has heavily relied on a few species of oxide transition metals with extremely high refractive indices[9–12,15], which are intrinsically difficult to achieve with organic molecules for example, proteins, lipids, carbohydrates and nucleic acids.

Here, we report Anderson light localization in biological nanostructures of native silk. We take inspiration from the 'brilliantly white', 'silvery' and 'lustrous' reflection of native silk fibres, possibly indicating strong light–matter interactions due to densely packed internal disordered nanostructures with reduced dimensionality. First, we image an individual silk fibre composed of numerous nanofibrils and their arrangement along the longitudinal axis of the silk fibre. Multiple imaging modalities, including scanning electron microscopy (SEM), transmission electron microscopy (TEM) and confocal microscopy, are used to visualize the relevant spatial information on the nanostructures of silk (e.g. size, distribution and dimension), minimizing sample artefacts. A nearly parallel nanofibril arrangement is highly advantageous for light interference and Anderson light localization, because it restricts a volume explored by light waves into the transverse plane as quasi-two-dimensional (2D) scattering. Second, although biological tissue is considered to be weakly scattering media, we predict that quasi-2D Anderson light localization can exist in the silk fibres with relatively modest refractive index contrasts. To realistically simulate the nanostructures of silk, high-resolution TEM images are post-processed to extract the complex shapes of the boundaries and domains of nanofibrils. Due to the geometrical complexity, we conduct a set of large-scale electromagnetic computations; the largest simulation contains ~$6 \times 10^4$ nanofibrils and ~$10^7$ triangular elements for the finite element method (FEM). Third, we observe that light transport is dominated by a few localized modes generated by scattering in the nanostructures of silk. When transmission matrix (TM) measurements visualize the locations of open transmission channels, their effective number (i.e. dimensionless conductance $g$) falls close to unity. Fourth, we validate the existence of localized modes by embedding moderate gain for exciting the cavities inside silk near the gain–loss equilibrium. The length scale of localized modes is measured using spatial scanning of the focused excitation beam and spectral mode decomposition. The statistics of decomposed modes differentiate silk from other diffusive structures, such as cellulose fibre microstructures of white paper. This approach is relatively insensitive to structural nonuniformity and light absorption of materials and limited numerical apertures (NA) in optics. Finally, we demonstrate that biogenic Anderson localization of light in native silk further gains a functionality of controlling radiation heat transfer. While light transmission is mostly suppressed due to Anderson light localization, giving rise to drastically enhanced reflectivity in the visible and near-infra-red (NIR) region, addition of high emissivity of the biomolecules of silk in infra-red (IR) radiation results in a significant radiative cooling effect during day and night in summer. Because Anderson light localization is a physical phenomenon not typically found in nature, our results from native silk may suggest an idea of 'natural' metamaterials such that nature overcomes constituent material limitations for exceptional properties, which are considered to be only possible by human engineering.

## Results

**Nanofibrillar structures in a silkworm silk fibre**. We image the size, distribution and dimension of nanostructures in native silk, using multiple imaging modalities. A silkworm secretes insoluble silk protein (i.e. fibroin) via numerous spigots of the spinneret (i.e. silk-spinning organ), which is crystallized to nanofibrils with voids and is assembled into fibroin filaments[16,17] (Fig. 1b, c; Supplementary Fig. 1). As condensed protein structures, a single silk fibroin filament (a silk fibre is composed of twin filaments, Supplementary Fig. 1b) with a size of $L \approx 20\,\mu m$ can contain ~3800 nanofibrils (Fig. 1d). The Fourier transform of nanofibrillar images (Fig. 1e) shows the absence of periodicity in the filament, free from a single spatial component (Fig. 1g). If each individual nanofibril serves as a scattering centre and the arrangement of scattering centres is favourable for constructive interference, Anderson localization could potentially be realized (Fig. 1a). The morphological characteristics inside a single silk filament obtained from the post-processed TEM image (Supplementary Fig. 1) provide indication for light localization. Although the size of a single nanofibril is ~25 nm (Fig. 1f), their natural distribution occasionally forms extended clusters, as shown in the large continuous dark granule-like areas (Fig. 1e). From an elastic light scattering standpoint, a single nanofibril or a cluster can be considered as a single scattering centre. In this case, the sizes of scattering centres $d$, ranging from 30 nm to 200 nm (size parameter $\pi d/\lambda \approx 1$ assuming $\lambda = 600$ nm and Fig. 1h), and the large number of scattering centres can enhance scattering power. These scattering centres are spatially and optically distinguishable (Fig. 1d, e, j), which agree with the reported characteristics of both silkworm and spider dragline silk with numerous parallel nanofibrils[18–25]. The volume fraction of nanofibrils (~30%; Supplementary Fig. 1) is close to an optimal value for maximizing the scattering power[26]. Moreover, the wave interference can be maximized by the distance between scattering centres of $131 \pm 37$ nm (~$\lambda/4$) (Fig. 1i), which allows the phase to be accumulated by ~$\pi/2$ in a silk filament. Thus, the morphology of silk appears to be optimized for rapid phase randomization as the light propagates, which is vital for decreasing the localization length of light $\xi$.

**Dimensionality of scattering process in native silk**. The dimensionality of structure and scattering process is a critical part of Anderson localization. In spite of the microscopically twisted appearance of silk fibres (Supplementary Fig. 2 and 3), the nanofibrils are nearly parallel along the longitudinal axis of the fibre and thus are not isotropically 3D. The reconstructed trajectories of nanofibrils show that nanofibrils are volumetrically continuous in 3D space and minimally intersect one another with a small local wedge angle variation of ~0.08° (Fig. 1j; Supplementary Movie 1). Although nanofibrils can have a worm-like packing pattern[27,28], involving the secondary structures of silk proteins (e.g. size of β-sheet nanocrystals of silk ≈ 2–12 nm[29,30]), the pattern exists on the nanoscale (« $\lambda$). The light perceives the worm-like packing pattern to be negligible, because the relevant size scale of structural variations is not comparable to $\lambda$. In Fig. 1j,

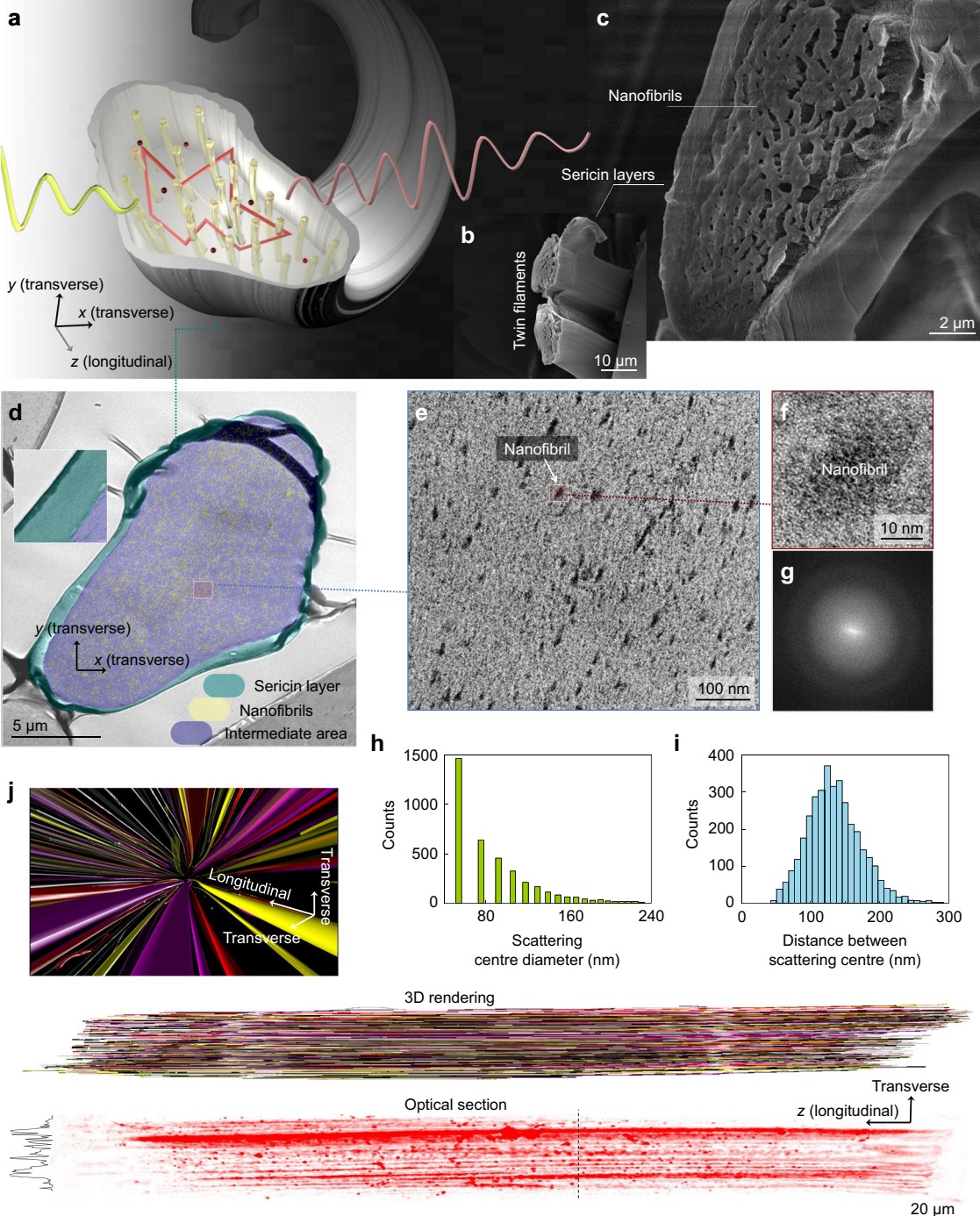

**Fig. 1** Nanofibrillar structures in a silkworm silk fibre. **a** 3D illustration of light localization in a single silk filament of nanofibrillar structures (yellow translucent cylinders). **b**, **c** SEM showing a freeze fractured edge of twin fibroin filaments **b** and nanofibrils **c**. **d**–**f** TEM with different magnifications. The colour-coded overlay in **d** distinguishes nanofibrils, voids and sericin layers. The dark granule-like dots in **e**, **f** indicate nanofibrils. **g** 2D fast Fourier transforms of **e**. **h**, **i** Histograms of scattering centre size (**h**) and distance (**i**). **j** 3D rendering of reconstructed nanofibril trajectories (cylinders) in a single filament (Supplementary Movie 1). Each connected component is labelled to a different colour. The 3D nanofibril trajectories are reconstructed from z-stack reflectance confocal microscopy images (Olympus FV1000) of a sericin-removed silk filament (×60 water immersion objective, confocal aperture of 55 μm, stack size of 7.5 μm and step size of 100 nm)

this high parallelism is preserved until the abrupt discontinuation of a filament due to microdefects[31]. In such a low-dimensional structure, a different localization mechanism is involved. The nearly paraxial (or parallel to the longitudinal axis) configuration can restrict a volume explored by the waves by minimizing the deviation of wave propagation from the transverse plane. In other words, the scattering strength is high in the transverse plane, while no significant scattering is present in the longitudinal direction. Thus, the process of light–matter interactions within a silk fibre can appropriately be treated as quasi-2D, while the silk cocoon shell is considered as an ensemble of such cavities of silk fibres. In a low-dimensional system (i.e. 1D and 2D), localization

is achieved once $L$ becomes greater than $\xi$; the Thouless number or the dimensionless conductance $g$ fall close to unity. In this case, the waves can essentially be localized in the limit of $L \to \infty$[32–36], following the random-matrix theory[37–41].

**Prediction of light localization**. To understand the critical role of nanofibrils in light localization, we consider a structure directly from the TEM micrograph (Fig. 1d; Supplementary Fig. 1). Even accounting for the low dimensionality of the silk fibre, the observation of localization is still challenged by an exponentially large $\xi$. For light confinement within an area over several micrometres, the constituent scattering materials require high refractive indices $n > 2$[42], which are not available in living organisms; the highest-refractive-index biological material is guanine only with $n = 1.83$[43]. However, using FEM (Methods), we find that the energy is localized exponentially within a single filament possessing the natural spatial distribution of nanofibrils with $n_{fibroin} \approx 1.5–1.6$[44] (Supplementary Methods). The confinement is evident from the high field intensity inside the filament (Fig. 2a), compared with the outside sericin layer, and from the exponentially decaying envelopes from the centre (Supplementary Fig. 4c, d). The local field width, which quantifies the degree of confinement (Supplementary Methods), is as narrow as 2.7 μm (Supplementary Fig. 4c). An average field width (~9 μm), which corresponds to $2\xi$, is smaller than the single filament size. On the other hand, the waves in a bare silk filament without nanofibrils (Fig. 2b) are highly coupled to the free-space modes, inducing leaky resonances; the average outgoing flux of the bare filament is ~1.52 times higher than that of the nanofibrillar filament (Fig. 2c, d).

As a statistical property of light localization in a finite system, $\xi$ is an intrinsic ensemble-averaged decay length of localized modes. From 100 different realizations of nanofibrillar structures, we calculate $\xi = 4.5$ μm (Supplementary Fig. 4a and Supplementary Methods), which is in good agreement with the average mode size of ~4.2 μm later measured by exciting the internal cavities near the gain–loss equilibrium. Interestingly, $\xi$ is shorter than the average filament size; $\xi < L$. Since the silk filaments can be stacked and cohesively bonded by the silkworm's spinning process, the total size of the filament cluster (i.e. cocoon shell) can significantly exceed the localization length $L \gg \xi$. Taking a cluster of filaments extracted from a TEM image into account, different arrangements of filaments are systematically included in simulations. As more filaments (yellow lines in Fig. 3b and white lines in Fig. 3c) are added to surround the initial localized area (blue lines in Fig. 3a), the field intensity outside the cluster (i.e. leaking radiation) disappears progressively, indicating that the wave becomes deeply localized. In this case, we analyse the underlying scattering process, by decomposing the wave components in the modes as a function of wavevector **k** from the Fourier transform of the field[35,45]. In Fig. 3d, e, the rings in **k**-space show that the waves are scattered into all possible directions with equal efficiency (or effective propagation constant). This result supports the idea that the localized modes in silk are isotropic. Owing to this isotropicity, the illumination from any angular directions can be coupled to the localized mode[36,46]. In **k**-space, the ring thickness implies the dispersion of the wave components, mediated by interference among multiple scattering paths, and the spread of the effective refractive index. Compared to the two-filament cluster (Fig. 3d), the interference in the 14-filament cluster is more drastic (Fig. 3e), significantly dispersing the spatial components (k-vector) of the scattered waves and slowing the propagation down; these indicate a transition from diffusive to localized regimes. Also, the intensity within the ring in **k**-space becomes more uniform such that the mode becomes more

isotropic. In other words, as the waves undergo more interference, the mode becomes more isotropic and deeply localized, away from the boundary, negligibly touching to the free space. Indeed, the average quality $Q$ factor in the 14-filament cluster has a 5.9-fold increase (Fig. 3f). Another consequence of the cooperation of multiple filaments is the enhancement of the mode density in wavelength (around $\lambda_0 = 600$ nm), which can allow for effective confinement of broadband light (e.g. sunlight) with an increased number of spectral peaks.

**Interrogation of localization by external illumination**. To characterize the wave localization experimentally, we directly obtain $g$ from transmission matrices (TM) (Fig. 4a and Methods). The dimensionless conductance $g$ is a fundamental localization parameter originally defined for the electronic wavefunction. For classical waves, the Landauer relation connects $g$ to an ensemble average of the transmittance $<T>$ such that $g = <T> = \Sigma_{b,a} T_{ba} = \Sigma_a T_a$, where $T_{ba}$ is the transmission intensity from an incident channel $a$ into an outgoing channel $b$ and $T_a = \Sigma_b T_{ba}$ is the total transmission[6,7,47,48]. When $g$ falls below or close to unity, the modes are separated in space and frequency (i.e. Anderson localization), showing sharp transmission peaks.

TM is obtained by optically measuring non-vanishing field speckle patterns $T_{ba}$ at the output plane (OP) (Fig. 4a) of the white silk cocoon (Supplementary Fig. 5a). Mimicking the incidence of sunlight from the free space onto the cocoon surface, a point-like source ($\lambda = 632.8$ nm) is illuminated on the surface of the silk specimen with an optical thickness of 14 ($\approx L/l_t$; Supplementary Methods) for imaging $T_{ba}$. For TM, $10^4$ images of $T_{ba}$ are assembled into a $100 \times 100$ array of $T_a(\alpha, \beta)$. Summing $T_a$ for all channels $a$, we directly obtain $g = 1.4$, which implies that localization in silk is proximal to the Anderson regime and that an effective number of open transmission channels is close to unity. This value is comparable with the state-of-art values of $g$; $g \approx 2.1–3.6$ in 3D GaP nanowires[15] and $g \approx 0.6–5.6$ in 2D ZnO nanoneedle arrays[49]. By comparing a similar synthetic structure, one can better understand the significance of the silk's nanoarchitecture. White paper is well known for its strong scattering with the white appearance from bleached cellulose fibre microstructures coated by kaolinite or calcium carbonate[50]; white paper is similar on the microscale yet different on the nanoscale, compared with the silk structures. Although the optical thickness of the paper specimen ($\approx 14$; Supplementary Methods) is comparable to the silk specimen, the light in the paper specimen is highly diffusive with an extremely high value of $g$ (Methods; Supplementary Fig. 6).

To account for a potential decrease in $g$ due to residual light absorption in silk, we also determine localization in a complementary manner, using the variance of $s_a$ ($=T_a/<T_a>$)[6,7]. $var(s_a)$ statistically measures the amount of fluctuations in the relative total transmission, which is the degree of intensity correlation of transmitted waves. In the entire area of $T_a(\alpha, \beta)$, the transport is dominated by the channel marked with $\Delta$ such that $T_a$ at $\Delta$ is significantly greater than $<T_a>$; the open transmission channels are visually countable and few in numbers (Fig. 4b). This results in a pronounced broadening of distribution function $P(s_a)$, deviating from a Gaussian distribution (inset in Fig. 4b)[51]. In turn, we obtain $var(s_a) = 0.5$, which is close to the critical value of Anderson localization transition of $2/3$[6,7]. In addition, the speckle pattern $s_{ba}(x, y)$ ($=T_{ba}/<T_{ba}>$) through the channel $\Delta$ has a peak with a FWHM of 0.8 μm and a height of 300 (Fig. 4c); the fluctuation in $P(s_{ba})$ is high, deviating from a Rayleigh distribution (inset in Fig. 4c). Such phenomena occur only when the light travels across nanofibrils through a small area on the fibre surface or the cohesively bonded stacks of silk fibres (Fig. 4f,

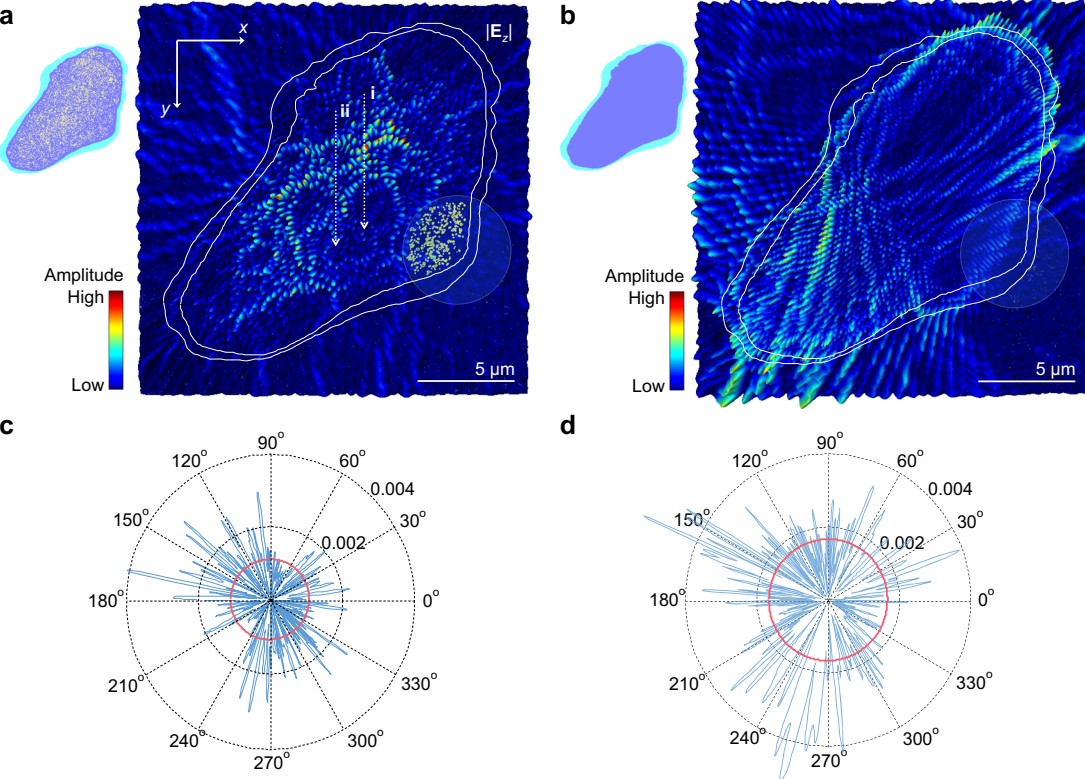

**Fig. 2** Riddle of nanofibrils. **a**, **b** Electric field amplitude |**E**$_z$| of modes in a single silk filament with (**a**) and without (**b**) nanofibrillar structures. The spreads of the fields along (i) and (ii) are 2.7 and 9 μm, respectively (Supplementary Fig. 4c, d). The overlays are the boundaries of sericin layers (white lines) and nanofibrils (yellow lines in inset of **a**) for computations. The boundaries and domains of nanofibrillar, interfibrillar and sericin areas are extracted from high-resolution TEM micrographs (Supplementary Fig. 1). **c**, **d** Far-field radiation patterns computed from the field of **a**, **b** using the near-to-far-field transformation, respectively. The red circles are average outgoing fluxes

g; Supplementary Fig. 7). For several incident channels $a$ around Δ, the patterns of $s_{ba}$ are surprisingly similar, giving rise to the horizontal bright line in the TM array (Fig. 4d), which is a tell-tale signal of the localized waves[52]; these identical speckles represent the pattern of one specific localized mode[36,46]. In contrast, where $T_a$ is suppressed, $s_{ba}$ is weakly correlated over different illumination positions (closed channels ◊ in Fig. 4b). Similarly to the channel marked with ◊ in the silk specimen, $s_{ba}(x, y)$ of the paper specimen is always decorrelated in the entire input plane (IP) (Supplementary Fig. 6c). Even in the potential presence of light absorption, the analyses of $s_a$ and $s_{ba}$ manifest the light localization in white silk, indicating light path crossing in the highly correlated speckle patterns[53]; the illumination is coupled to a small number of localized modes.

**Interrogation of localized modes by internal luminescence**. To rule out any suspicion on whether we observe Anderson localization or not, we further adapt a different type of experimental scheme, which is technically free from the limited NA in optics, the nonuniformity of sample structures and the light absorption of sample materials. We excite cavities located far away from the sample surface using internal light sources and quantify the cavities in multiple dimensions of space **r**, frequency $\omega$ and excitation energy $E_{ex}$. Embedding moderate gain (Supplementary Fig. 8) is a particularly useful way for examining the affluent phase space of localization without modifying the resonant properties of the underlying passive cavities, because nonlinear effects of gain saturation and mode competition are almost negligible for localized specimens[13,42,54–57]. At around the gain–loss equilibrium (i.e. lasing threshold), the system compensates the

intrinsic loss and resembles a conservative medium (e.g. electronic conductors), in which the time-evolution operator is Hermitian. In this case, the electromagnetic emission can be expressed by a complete set of quasinormal modes[7,13]:

$$I_m(\mathbf{r}, E_{ex}, \omega) = \sum_{h=1}^{H} a_h(\mathbf{r}, E_{ex}) \frac{\Gamma_h(\mathbf{r}, E_{ex})/2}{i(\omega - \omega_h(\mathbf{r}, E_{ex})) + \Gamma_h(\mathbf{r}, E_{ex})/2},$$
(1)

where $\omega_h$ ($= 2\pi c/\lambda_h$), $\Gamma_h$ and $a_h(\mathbf{r}, E_{ex})$ are the central frequency, the linewidth and the intensity of $h$th mode in the space-energy domain of $\mathbf{r}$–$E_{ex}$. We note that this quasinormal mode description has been employed for light transport in passive systems[7]. To estimate the set of the unknowns (i.e. $\omega_h$ and $\Gamma_h$), the mode expansion in Eq. (1) is fitted to the measured spectra $I_e$ in frequency (Fig. 5a; Supplementary Fig. 9 and 10). In $I_e$ ($E_{ex}, \lambda$), the decomposed modes have stable and non-interacting central wavelengths $\lambda_h$ (Fig. 5b) and their intensity grows linearly one at a time (Fig. 5c). In Fig. 5b, raster scanning of the specimen at around the gain–loss equilibrium (=3.5 μJ/pulse) enables to reveal a plethora of resonances in $I_e(\mathbf{r}, \lambda)$, which are sharply confined in space and separated in wavelength (i.e. mode spacing ≥ linewidth). In the absence of nanofibrils, those sharp features, however, disappear (Fig. 5a; Supplementary Fig. 11), as commonly observed in diffusive lasing. Clustering the modes that share the identical $\omega_h$ and $\Gamma_h$ (marked with the same colours of yellow–pink in Fig. 5b)[13,58], we directly measure the spatial extent of exponentially decaying intensity profiles (or mode length) ~4.2 μm (Fig. 6f and Methods). These suppressed modal interactions and small mode volumes are the hallmarks of Anderson-localized modes[13,55].

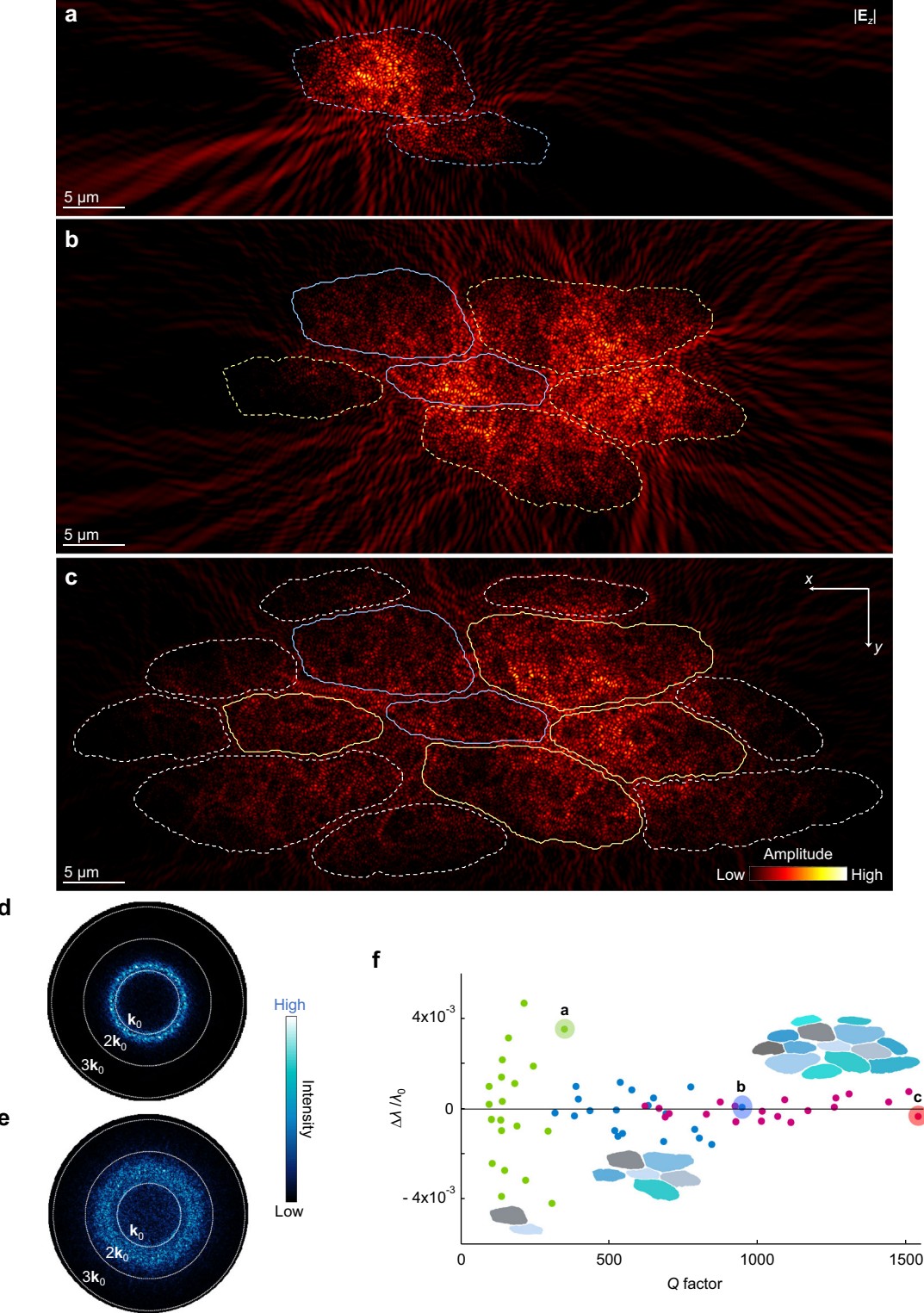

**Fig. 3** Prediction of light localization. **a**–**c** Electric field amplitude |$\mathbf{E}_z$| of modes in 2-cohesively (**a**), 6-cohesively (**b**) and 14-cohesively (**c**) bonded stacks of silk filaments (with nanofibrillar structures) at the wavelengths marked by the larger circles in **f**. The inner boundaries of sericin layers are marked by the coloured lines. **d**, **e** Norms of the Fourier transform of the electric field $\mathbf{E}_z$ of **a**, **c**, respectively. **f** $Q$ factors of adjacent 20 modes around $\lambda_0 = 600$ nm. The green, blue and red dots correspond to the  2-filament, 6-filament and 14-filament structures, respectively

We further analyse the spatiotemporal properties of the decomposed modes and their $E_{ex}$ dependence. In the nanofibrillar structures (i.e. silk), the light is confined in an area smaller than a single filament (Figs. 5d and 6f), while in the absence of nanofibrils the light is pronouncedly diffusive (Fig. 5e; Supplementary Fig. 11e, f). In the silk specimen, the small mode size leads to an enhancement of the electromagnetic field and enables gain to compensate the loss at low $E_{ex}$. Compared to the paper

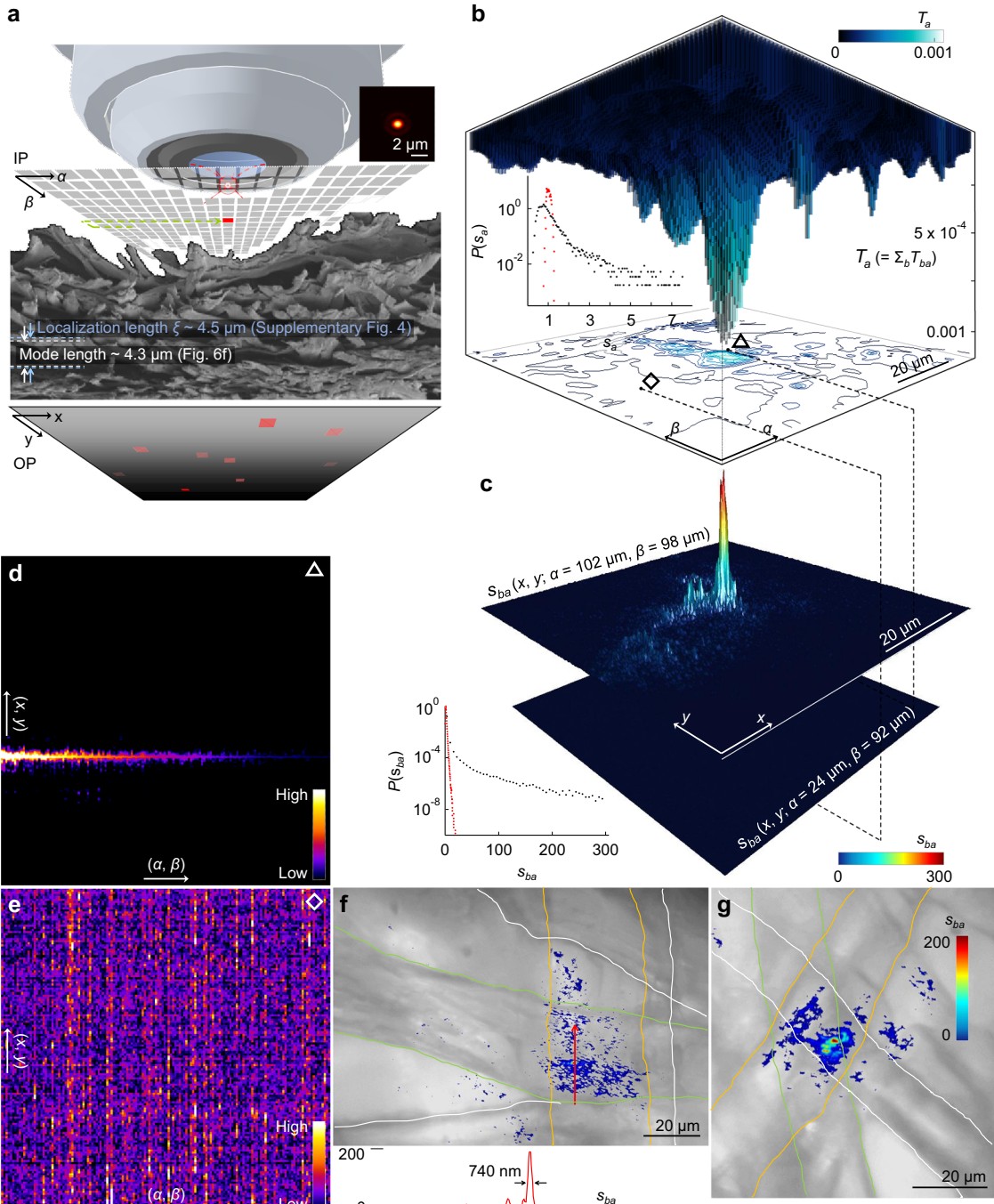

**Fig. 4** Interrogation of localization by external illumination. **a** Schematic of optical transmission matrix (TM) measurements. External point-like illumination (inset: focal spot with a diameter ~1 μm) is scanned on the array of the input $\alpha$–$\beta$ plane (IP) and non-vanishing optical intensity through modes is imaged on the output $x$–$y$ plane (OP). **b** $T_a(\alpha, \beta)$ map of the silk cocoon. The bottom projection image visualizes a small number of transmissive channels. Inset: probability distributions of $s_a$ for the cocoon specimen (black dots and var$(s_a) = 0.5$) and the paper specimen (red dots and var$(s_a) = 5.2 \times 10^{-3}$). **c** $s_{ba}(x, y)$ maps from two locations of $\triangle$ and $\diamondsuit$ in **b**. Inset: probability distributions of $s_{ba}$ from $10^4$ translations of illumination for the cocoon (black dots) and paper (red dots) specimens. **d**, **e** TMs around two locations of $\triangle$ and $\diamondsuit$ in **b**. Each column of TM is composed of $T_{ba}(x, y)$, recorded at a position of $(\alpha, \beta)$. **f**, **g** $s_{ba}(x, y)$ overlay (pseudo-colour) on white-light microscopy images (greyscale). Fluctuations with a sharp field peak (FWHM ~740 nm) are located on the stacks of multiple silk fibres (Supplementary Fig. 7). The boundaries of filaments are marked by the coloured lines

specimen, the conversion efficiency ($=dI_e/dE_{ex}$) has a 4-fold increase and the lasing threshold has a 2.5-fold decrease (Figs. 5f, g and 6a–c). The asymmetric shape of the silk's threshold distribution (Fig. 6c) also indicates that the light undergoes strong scattering[59]. Indeed, the $Q$ factors ($= \omega_h/\Gamma_h$ and inversely proportional to the energy decay rate of the mode) range from 2000 to 8000 (Fig. 6d), which are comparable to the values from

single quantized emitters in photonic crystal waveguides[60]. We obtain the cavity storage time by the Fourier transform of the frequency variation term in Eq. (1) for $t > 0$. The cavity storage time of the silk fibres is ~130 times longer than that of the bare fibres (Fig. 6e) and is estimated to be ~6.5 times longer than that of 3D white beetle scales with brilliant whiteness[61,62]. Finally, we repeat the statistical analyses for $P(I_e/<I_e>)$. As $E_{ex}$ increases,

the system results in larger fluctuations in $I_e(\mathbf{r})$ (Fig. 6g) recovering the light confinement and the intensity variance var $(I_e / <I_e>)$ increases monotonically around the threshold (i.e. $E_{ex}$ = 3.0 µJ/pulse) to surpass >2/3 (Fig. 6h), which is the onset of

Anderson localization[6,7]. We note that some modes are not loss-compensated and are damped at $E_{ex}$ = 3.0 µJ/pulse, resulting in slightly lower var$(I_e / <I_e>)$ (marked with + in Fig. 6h) than that of the passive system (=0.5 and Fig. 4b). The peculiarity of our

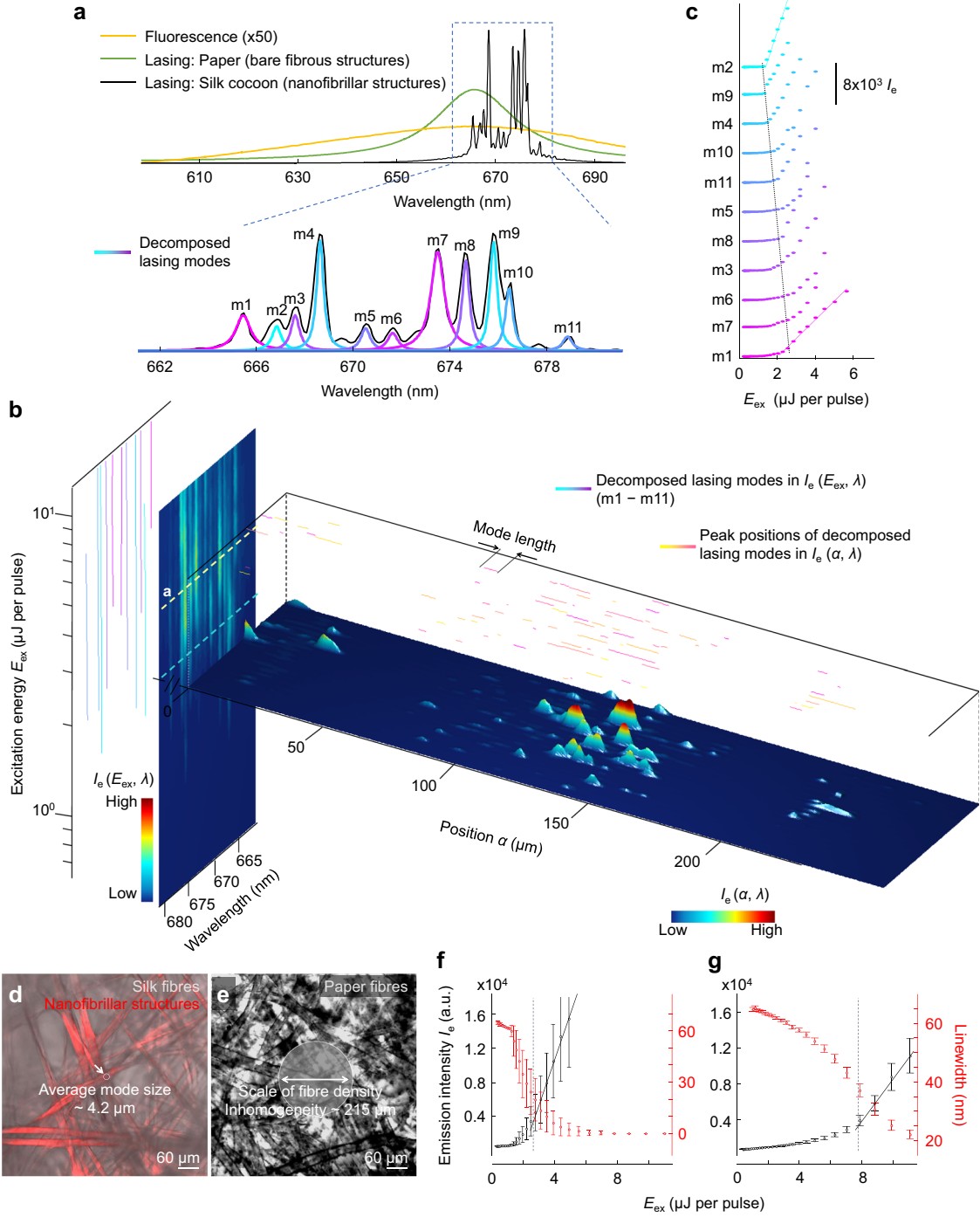

**Fig. 5** Interrogation of localized modes by internal luminescence. **a** Modal decomposition (blue–magenta) of transmitted emission $I_e$ (grey line) in **b** by inversely estimating $a_h$, $\omega_h$ and $\Gamma_h$ in the analytical expression of Eq. (1). The modes (m1–m11) are colour-coded with the thresholds plotted in **c**. **b** Space-energy wavelength spectrogram of emission $I_e(\mathbf{r}, E_{ex}, \lambda)$. Horizontal plane: $I_e(\mathbf{r}, \lambda)$ near the gain–loss equilibrium (=3.5 µJ/pulse). $\mathbf{r}$ is the spatial coordinate on the input $\alpha$–$\beta$ plane (IP). $\lambda_h$ and $\Gamma_h$ are clearly identified over the position $\alpha$ (on IP), where the mode intensity is greater than the baseline. The $\lambda_h$ lines on the top (yellow–pink) delineate the spatial extent of modes. Vertical plane: $I_e(E_{ex}, \lambda)$. The $\lambda_h$ lines on the left (blue–magenta) show suppression of modal interactions. **c** $a_h$ of single lasing modes decomposed in **a** as a function of $E_{ex}$. The blue and magenta colours indicate the lowest and the highest threshold. **d, e** Average size of modes in silk **d** and average length scale of spatial intensity fluctuations in paper **e**, overlaid on confocal microscopy images. **f, g** Average intensity and linewidth of the decomposed lasing modes in silk **f** and the diffusive lasing peaks in paper **g** as a function of $E_{ex}$. The vertical yellow lines mark the average $E_{ex}$, corresponding to the system gain–loss equilibrium. The error bars are the standard deviations of the intensity and linewidth from 259 lasing modes shown in Supplementary Fig. 10

observation is clear, when comparing to the Gaussian distributions from the typical fibrous microstructures (e.g. white paper) even at high $E_{ex} > 10\,\mu J/pulse$. This result serves as the evidence that we observe Anderson-localized modes in silk near the gain–loss equilibrium, making fundamental differences in light–matter interactions for the topologically similar (on the microscale) yet different (on the nanoscale) structures of white paper.

**Biogenic light localization reveals a self-cooling function of native silk.** Anderson localization suppresses most of transmitted light and gives rise to the drastically enhanced reflectivity in the visible and NIR region (left panel of Fig. 7d); native silk shows the

brilliant whiteness where solar radiation dominates. When the strong reflectivity in visible/NIR is supported by the biomolecules of silk with high emissivity in the IR region ($\lambda = 2$–$20\,\mu m$), native silk can satisfy the stringent requirements of passive radiative cooling (i.e. high reflectivity in visible/NIR and high emissivity in IR)[63–65], as the surface radiates heat to outer space more than it absorbs. In Fig. 7d, the inner and outer compartments of a silk cocoon with a thickness of $400\,\mu m$ reflect 90% of sunlight at the peak of AM1.5 spectrum and 80% on average for $\lambda = 0.4$–$1\,\mu m$. This ensemble average reflectance is higher than the reported reflectance for other biological systems[61,62] and is even compatible with the reflectance of a metal reflector (e.g. aluminium). In the IR region, radiation energy is strongly transferred or absorbed

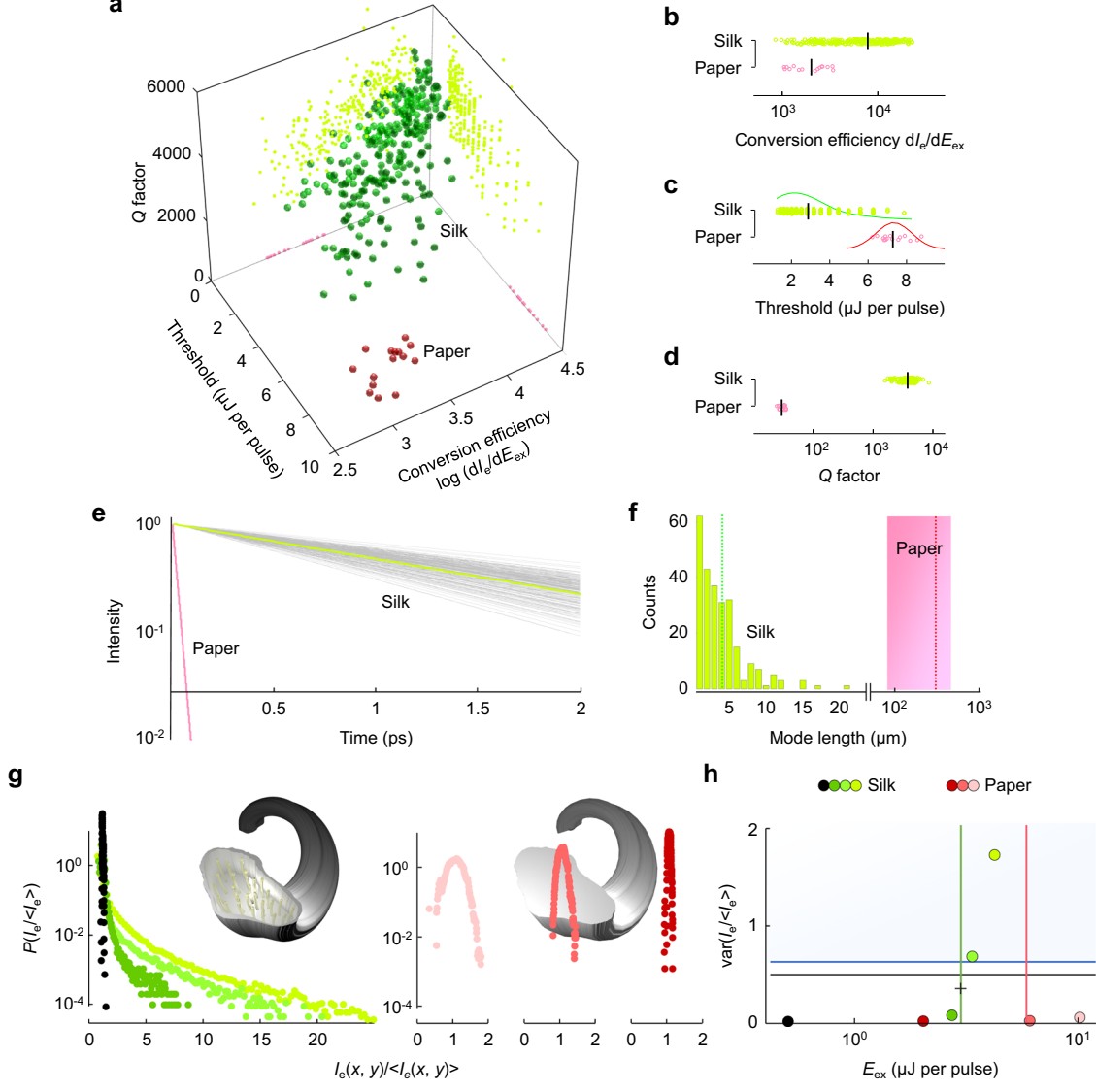

**Fig. 6** Statistical evidence on localization. **a** $Q$ factor, threshold and conversion efficiency of decomposed modes. The $Q$ factors are evaluated at the gain–loss equilibrium (Supplementary Fig. 10). The axial-plane projections show that the modes of high $Q$ factors have the enhanced lasing performance. The nanofibrillar structures (green dots) and the bare microfibrous structures (red dots) have the distinct properties of resonances and lasing. **b**–**d** Pairwise comparisons of the distributions of the $Q$ factor **b**, the threshold **c** and the conversion efficacy **d** for silk and paper. **e** Time evolution of energy decay for the decomposed modes. The average decay rate of $1.38\,ps^{-1}$ (green line) fundamentally corresponds to an average $\Gamma_h$ of $0.11\,THz$. **f** Mode lengths in silk determined from $I_e(\mathbf{r}, \lambda)$ (Methods). The red box marks a range of the spatial intensity fluctuations in paper. The dotted vertical lines mark their average sizes. **g** $P(I_e / < I_e>)$ for silk (left) and paper (right) at different $E_{ex}$ that is colour-coded in **h**. **h** Degrees of localization captured by $\text{var}(I_e / < I_e>)$ at different $E_{ex}$. The blue area indicates the Anderson localization regime above 2/3. The black line indicates $\text{var}(s_a)$ of the passive systems (Fig. 4b). The + symbol estimates $\text{var}(I_e / < I_e>)$ at the lasing threshold in silk (green vertical line). The vertical lines are the average thresholds of each specimen (Fig. 5f, g)

into the biomolecules through vibrational transitions, because the biomolecules are composed of highly polarized vibrating bonds. At the thermodynamic equilibrium, such high absorption mediates high thermal emission by Kirchhoff's identity[66].

Native silk has a unique emissivity profile in IR, resulting from the endogenous vibrational transition of the silk protein structures. In the right panel of Fig. 7d and Supplementary Fig. 12d, the series of vibrational modes in the silk protein structures from *Bombyx mori* (e.g. silk fibroin) overlap and are excited together, resembling beads in a necklace. In this case, the strong absorption is mainly attributed by the protein backbone (i.e. amide group) in the primary structure of fibroin protein[67,68], which is a repetitive amino acid sequence (Gly-Ser-Gly-Ala-Gly-Ala)$_n$. Depending on its secondary structures (e.g. β-sheet, random coil and α-helix), the absorption peaks are slightly shifted, which in turn increase the linewidth of amide bands. We note that the antiparallel β-sheet secondary structure is predominant in fibroin protein and is mainly responsible for the IR absorption as well as the mechanical properties of silk

fibres[16,30,69]. Also, the bands of amino acid chains add non-zero IR absorption adjacent to the strong amide bands. This super-broad absorption of silk proteins extends from amide I (6 μm) to amide V bands (12–20 μm) with an average emissivity of 0.87 covering the first and the second atmospheric transparency windows (i.e. 7–14 μm and 16–25 μm). In contrast, typical synthetic emissive materials (e.g. silicon dioxide ($SiO_2$) or hafnium dioxide ($HfO_2$)) have single-peaked narrow absorption with a FWHM of 2–5 μm[63,65]. Thus, upon access to a clear sky (through the atmospheric transparency windows in Fig. 7d), native silk can effectively dissipate heat. Using a regenerated silk film, we further characterize the fundamental dielectric response of silk protein by measuring the real and imaginary parts Re($n$) and Im($n$) of complex refractive index at all of the wavelengths in the visible/NIR/IR regions (Fig. 7e; Supplementary Methods). Im($n$) of silk further confirms the critical role of amide groups on the resonant absorptive processes upon IR radiation. As shown in the inset in Fig. 7e, out-of-plane molecular motions take part in longer vibrational wavelengths, while in-plane bending δ and

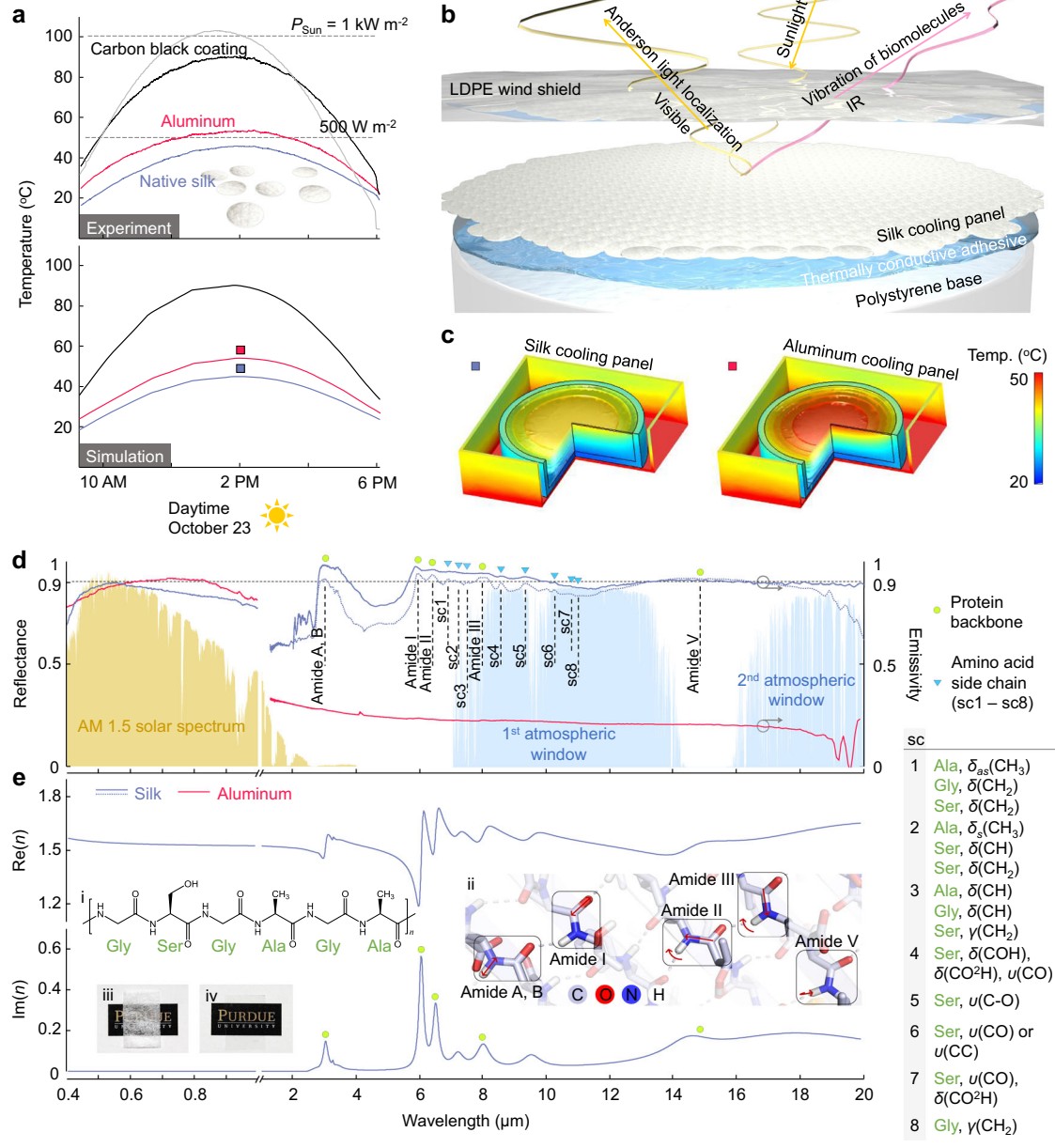

stretching $\gamma$ are responsible for shorter vibrational wavelengths. Naturally, silkworm silk possesses an integrated property of a solar reflector and a thermal emitter, which are formidable to coexist in a single synthetic (or man-made) material.

With an experimental configuration that selectively accounts for radiative heat exchange (Methods; Supplementary Fig. 12), we discover that native white silk keeps cool during day and night in late summer. When the solar power is as high as 1000–1030 W m$^{-2}$, a 2D assembly of silk cocoon patches reduces the steady-state temperatures in a range from 42 to 44 °C (Fig. 7a); under the identical solar power, carbon black paint heats up nearly 90 °C. To be exposed with the maximum solar power of 1030 W m$^{-2}$, the outdoor testbeds are tilted 50° to south, accounting for the solar elevation angle of the measurement days in West Lafayette, Indiana. By comparing a conventional reflecting material (i.e. aluminium), one can better capture the significance of native silk's cooling performance. At the peak sunlight (2 p.m. local time), the temperature of silk panel drops 10 °C below the temperature of the aluminium panel, which can be sufficient to avoid overheat above a lethal thermotolerance of 45 °C for silkworms[70]. Moreover, we theoretically analyse the interplay between the sunlight reflection/thermal emission and radiative cooling, using 3D heat transfer models of cooling panels that consider radiative, convective and conductive heat transfer. The models take into account the exact 3D geometry of the testbeds placed in the backyard (Supplementary Fig. 12a), incorporating the emissivity data in Fig. 7d, the solar power data, the ambient temperature data and the thermal properties of the components (Methods). The predicted steady-state temperatures are in good agreement with the outdoor measurements under the clear sky (bottom of Fig. 7a). In Fig. 7c, the temperature distribution visualizes the significant temperature drops at immediately below the back surface of silk compared to aluminium, while the acrylic boxes enclosing the silk and aluminium panels have the same temperature distributions. Given the similar level of solar absorption in silk and aluminium (i.e. similar reflectance values in Fig. 7d), the enhanced heat dissipation in silk is attributed to the intrinsic thermal emission of biomolecules. In this case, night-time measurements provide useful information, because the gain terms of radiative heat source in the 3D heat equation become negligible by disappearance of sunlight and the cooling effect at night under a clear sky is purely determined by the emissivity in IR (Methods). As the Sun sets beyond the horizon (6 p.m. local time), the temperature curve of silk converges to that of carbon black paint, which has an emissivity close to unity mimicking a black body. This result shows that the high emissivity of biomolecules indeed contributes to the enhanced heat dissipation (Supplementary Fig. 13); the temperature of aluminium has a 0.8 °C constant positive offset from both silk and black paint. Among several functions as a habitat for regulating physical and biological conditions[71–74], radiative cooling would be vital for the survival and reproduction of silkworms, since the pupae of silkworms spend the most important period in their life inside the silk cocoon[70]. From an animal behaviour standpoint, the silkworm's effort for avoiding heat retention is usually observed in behaviour to spin its cocoon in a low-light environment (e.g. shadow of leaves). By virtue of Anderson light localization, native silk itself is highly reflective in visible/NIR in conjunction with the highly vibrational nature in IR, all of which in turn keep cool under extreme sunlight and low wind conditions. We note that native silk (e.g. cocoons and fibres) could easily be mass-produced and allows large-scale continuous integrations (Fig. 7b; Supplementary Fig. 12a). Thus, native silk will be one of the economical renewable energy options by offering an opportunity to enable electricity-free biophotonic cooling in off-grid areas.

## Discussion

The theories of Anderson light localization have been extended to the realm of biological and natural systems. Our findings will be an ab initio foundation for opening up largely unexplored opportunities for utilizing biogenic resonances. This purely physical phenomenon has biological relevance and can be applied to engineering, energy and biomedical areas. Based on the bio-compatibility and bioresorbability[44], silk can provide a new solution for the current critical demand on implantable optical components for in vivo optical imaging (e.g. brain-machine interface) without adverse effects. Taking advantage of the transverse Anderson localization of light, image information could potentially be transported along the longitudinal axis of a silk fibre. In this case, the obvious next step would be to adapt a transverse localization scheme[34,75,76], in which illumination is coupled into the transverse dimension of a single silk fibre, so as to observe purely localized waves that stay after diffusive propagation, excluding leakage from the input. Enhanced light–matter interactions of Anderson light localization can be applied to highly sensitive optical biosensors. Accounting for high $Q$ factors of localized modes, silk can response to subtle changes in the refractive index of medium or to target molecules by attaching recognition receptors. Thus, we envision a new class of biosensing platforms, which are wearable and flexible as textiles, for prolonged monitoring of physiological and biological changes in body fluid (e.g. sweat, urine and saliva). Our findings may also change the current paradigm of designing light-based therapies through biological hard tissue (e.g. sclera, bone, teeth and cuticle). As an alternative therapeutic optical window where light penetrates deep into biological tissue, localized modes could potentially be utilized to deliver light energy. Energy can also be confined in a small area by strong resonances, minimizing damage around the target tissue causing pain, swelling and scarring. As wave localization is universal, this idea can further be

**Fig. 7** Self-cooling function of native silk revealed by biogenic light localization. **a** Top: measured steady-state temperature of native silk, a conventional reflecting material (i.e. aluminium) and a high-emissive material (i.e. carbon black paint-coated aluminium) under direct sunlight exceeding 1000 W m$^{-2}$ (grey line) on a clear day of late summer in West Lafayette, Indiana. The temperature is measured at the centre immediately below the back surface of each cooling panel. Bottom: computed steady-state temperature using a 3D multiphysics model of heat transfer (Methods). **b** 3D illustration of the scalable biophotonic cooling device, continuously integrating unit silk patches with a diameter of 10 mm. Photograph of the device is shown in Supplementary Fig. 12a. **c** Computed temperature distributions below the silk (left) and the aluminium (right) cooling panels at the solar irradiance of 1000 W m$^{-2}$ (2 p.m. local time). **d** Ensemble-averaged reflectance in the visible/NIR region and emissivity in the IR region from the inner/outer compartments of an white silk cocoon with a thickness of ~400 μm (blue solid lines). For the entire solid angle $\Omega$ collection, two hemispherical measurement systems are used for the visible/NIR (Thorlabs IS200-4) and IR (Surface Optics SOC-100 HDR) regions. After loss compensation for the limited NA of a Fourier transform infra-red (FTIR) microscope (Supplementary Methods), FTIR microscopy measurements (blue dotted line) further provide enhanced visibility of the vibrational modes of silk protein marked with ○ and Δ; sample thickness ~100 μm. In the list of vibrational modes, the label prefixes $\delta$, $\gamma$ and $\nu$ indicate bending, rocking and stretching vibrations. The suffixes $as$ and $s$ imply asymmetric and symmetric modes. **e** Real and imaginary parts Re(n) (top) and Im(n) (bottom) of the complex refractive index of a regenerated silk film in the entire visible/NIR/IR regions. Amide bond links amino acid monomers in fibroin protein (i) and has five vibrational modes within the wavelength of interest (ii). In spite of the identical protein composition, the colour and scattering of silk drastically change if native silk (iii) loses the light localization structure to form the regenerated film (iv)

extended to the existing clinical technologies using mechanical waves (e.g. high intensity focused ultrasound[77]).

Counterintuitively, we have shown that the phenotypic structures of silk produced by silkworms (*Bombyx mori*) bring their natural protein structures into the regime proximal to Anderson localization. Overall, the presented results will broaden our understanding of linear wave phenomena in biological and natural systems with strong interference and resonances, not just in optics.

## Methods

**Numerical experiments of modes using FEM.** The critical information on transport is carried in modes, which can be obtained by solving the Hamiltonian of the system. To compute optical modes (also known as quasibound resonances) in a disordered medium for light scattering problems, we considered a discretized Helmholtz equation that can be numerically solved by FEM[42,78]. For realistic simulations of a single silk filament and multiple filaments in a silk cocoon shell, we extracted the boundaries and domains of nanofibrils, interfibrillar areas and sericin layers after post-processing high-resolution TEM micrographs (Supplementary Fig. 1a). Each silk filament typically contains $2-6 \times 10^3$ nanofibrils, depending on the filament diameter. Thus, we conducted a series of large-scale electromagnetic simulations containing up to $\sim 6 \times 10^4$ nanofibrils (in 14-filament structures) and generating $\sim 10^7$ triangular elements for FEM computations. The experimental values of refractive indices $n$ of nanofibrils, interfibrillar areas and sericin were 1.6, 1.0 and 1.34 at $\lambda = 600$ nm, respectively[44] (Supplementary Methods). For a uniform single silk fibre without nanofibrils, the refractive index of interfibrillar areas was set to be 1.6 to make zero refractive index contrast between the nanofibril and interfibrillar areas. The maximal mesh sizes were set to $\lambda/10$ in the interfibrillar areas and were reduced to $\lambda/20$ inside the nanofibrils due to their complex shapes (Supplementary Fig. 1d). For the boundary condition, a scattering boundary condition was applied on the outer computation domain up to 68 μm × 27 μm. Using RF Module of COMSOL Multiphysics (4.3a version), we computed all of the modes near $\lambda_0 = 600$ nm for each disordered system and displayed the norm of $\mathbf{E}_z$ field component of the transverse magnetic mode. Each mode is represented by eigenfrequency $\kappa$, of which the real part $Re(\kappa)$ is a resonant frequency and the imaginary part $Im(\kappa)$ is associated with the width of resonance that determines the leaking radiation out of the system (i.e. loss). This yields the quality $Q$ factor defined as $Q = -Re(\kappa)/2Im(\kappa)$. We analysed the field of the modes to characterize the angular leaking radiation intensity (Fig. 2c, d) and to decompose wave components as a function of wavenumber $k = 2\pi/\lambda$ (Fig. 3d, e).

**Optical measurements of transmission matrices.** To characterize light localization in the white silk cocoon, the inner compartment of a native silk cocoon of *Bombyx mori* (Baekokjam) ($L = 220$ μm) was used as experimental specimens; the fibre distribution of the inner compartment is more uniform than that the outer compartment (Supplementary Fig. 3c). As shown in Fig. 4a, we obtained TMs by imaging optical speckle patterns $T_{ba}$ through the specimen at the output plane (OP) $(x, y)$ using a microscopy imaging setup[15,79,80]. A collimated beam from a HeNe laser ($\lambda = 632.8$ nm) was focused on the input plane (IP) with a FWHM of $\sim 1$ μm (inset of Fig. 4a) via an objective lens with NA = 0.9 (Meiji MA970 Plan Achromatic) for point-like illumination. The transmission intensity at OP was collected via another objective lens with NA = 0.9 (Meiji MA970 Plan Achromatic) and was imaged using a four-megapixel interline monochrome CCD camera. Although the optically limited NA (<1) corresponds to a loss of channel control, which changes the probability distribution of transmission eigenvalues, it does not substantially modify the distribution of transmission intensity $T_{ba}$ (Fig. 4c)[81]. The point-like illumination was scanned on channels at the transverse dimension of IP $(\alpha, \beta)$, by moving the specimen mounted on a two-axis motorized micropositioner (Zaber T-LSR150A). Taking the full width of illumination into account, the unit translation step in $(\alpha, \beta)$ positions was set to 2 μm to avoid overlapping illumination between adjacent positions. For each TM, $10^4$ images of $T_{ba}$ were acquired by translating the specimen in an area of 200 μm × 200 μm, which results in a $100 \times 100$ array of $T_a$. TM provides a full account of the input–output responses[42,80,82] (i.e. $T_{ba}$, $T_a$ and $T$) and the statistics of their normalizations (i.e. $s_a = T_a/<T_a>$ and $s_{ba} = T_{ba}/<T_{ba}>$), making it possible to directly assess $g$ in a single system. To minimize the effect of the background intensity on statistical analyses of $P(s_{ba})$ and $P(s_a)$, we used image region defined by a half maximum of the ensemble-averaged image. Distributions of $s_{ba}$ and $s_a$ were compared with diffusive specimens of white paper ($L = 100$ μm, $L/l_t \approx 14$, $<\cos \theta> = 0.1$; Supplementary Fig. 5b), which also has fibrous microstructures (Supplementary Fig. 6).

**Measurements of $I_e(r, E_{ex}, \lambda)$ and analyses of localized modes.** To confirm the existence of localized modes in silk, we embedded luminescence sources to couple into internal cavities, which can be transferred to far-field radiation patterns near the gain–loss equilibrium. In sericin-removed silk filaments[83], the 3D volume of nanofibrillar structures was uniformly infiltrated by DCM in dimethyl sulfoxide (low reabsorption and quantum yield of 75%) at a low concentration of 0.5 mg/ml

(Supplementary Fig. 8). An excitation beam from a frequency-doubled Q-switched Nd:YAG laser (pulse duration of 400 ps, repetition rate of 5 Hz and $\lambda$ of 532 nm) was focused on the specimen with a FWHM of $\sim 1.8$ μm via an objective lens with NA = 0.4[56,57,84]. $I_e(\mathbf{r}, E_{ex}, \lambda)$ was collected via an objective lens with NA = 0.9 used in the TM setup and was detected by a spectrometer with a spectral resolution of $\sim 0.15$ nm, while scanning the specimen with a unit translation step of 500 nm and controlling $E_{ex}$ using a continuous variable linear neutral density filter. For simultaneous modal decomposition, we searched $\omega_h$, $\Gamma_h$ and $a_h$ that minimize the sum of squared differences between Eq. (1) and the measured spectra $I_e$, using the Nelder–Mead simplex method[85,86]:

$$\min F = \sqrt{\frac{1}{S}\sum_{s=1}^{S} \|I_e(\omega_s) - I_m(\omega_s, a_h, \omega_h, \Gamma_h)\|^2}, \quad (2)$$

where $F$ is the objective function to be minimized, $\|...\|$ is the Euclidian norm, and $S$ is the number of points within the frequency of interest. An average number of modes per $\mathbf{r}$ and $E_{ex}$ was $H = 12$ with 36 unknowns. Because the iterative solutions can be trapped in local minima or can be diverged in such multivariate optimization, the algorithm was set to contain multiple groups of initial guesses[85,86]. A representative example of the fitted result $I_m(E_{ex}, \lambda)$ is shown in Supplementary Fig. 9 and the full results of decompositions (i.e. $a_h$, $\omega_h$ and $\Gamma_h$) are visualized in Supplementary Fig. 11a–c. We evaluated the $Q$ factors ($=\omega_h/\Gamma_h$) at the threshold of each decomposed mode. In $I_e(\mathbf{r}, \lambda)$, the mode length $l_m$ was obtained:

$$l_m = \frac{\int \varepsilon(\mathbf{r}) I_m(\mathbf{r}) d\mathbf{r}}{\varepsilon(\mathbf{r}_0) I_m(\mathbf{r}_0)}, \quad (3)$$

where $I_m(\mathbf{r})$ is the spatial emission profile of the decomposed mode sharing the same $\omega_h$ and $\Gamma_h$, $\varepsilon$ is the dielectric constant, and $r_0$ is the anti-node position of the mode. Finally, single-shot measurements over 300 successive excitation pulses were used to verify the temporal stability of lasing spectra used for the current mode analyses (Supplementary Fig. 10d, e).

**3D multiphysics modelling of heat transfer.** The critical function of self-cooling in native silk is regulated by the sunlight reflection and the thermal radiation, of which their spectral behaviour is distinctly separated into the visible/NIR and IR regions, respectively. To compute a steady-state solution of temperature $T$ for radiative heat exchange with convection and conduction, we combined the terms of radiative heat source $Q_r$ and convective heat source $Q_c$ with a 3D heat boundary value problem:

$$\rho C_p \frac{\partial T}{\partial t} + \nabla \cdot (-k_c \nabla T(x, y, z)) = Q_r + Q_c$$
$$= \int_{4\pi} \int_0^\infty \varepsilon(\lambda, \Omega)(R_{sun}(\lambda, \Omega) + R_{atm}(T_{amb}, \lambda, \Omega) + R_m(T, \lambda, \Omega) - R_{bb}(T, \lambda, \Omega)) d\lambda d\Omega$$
$$+ h_c(T_{amb} - T) \text{ for } (x, y, z) \in \Pi, \quad (4)$$

where $\rho$ is the solid density, $C_p$ is the heat capacity, $\nabla \cdot \nabla = \frac{\partial^2}{\partial x^2} + \frac{\partial^2}{\partial y^2} + \frac{\partial^2}{\partial z^2}$ is the Laplace operator, $k_c$ is the thermal conductivity, $\varepsilon$ is the emissivity (Fig. 7d), $h_c$ is the convective heat transfer coefficient, $T_{amb}$ is the far distant air temperature in the direction of exposure, and $\Pi$ is the geometry domain. Then, we defined the boundary condition on the cooling panel using the flux of convective heat:

$$\mathbf{n} \cdot (k\nabla T(x, y, z)) = h_c(T_{amb} - T(x, y, z)) \text{ on } (x, y, z) \in \partial^{testbed}\Pi \quad (5)$$

where $\mathbf{n}$ is the normal vector toward exterior. In Eq. (4), $Q_r$ is balanced by the cumulative and dissipative contributions from $R_{sun}$, $R_{atm}$, $R_m$ and $R_{bb}$, which are the solar radiation, the atmospheric radiation, the mutual radiation with other objects and the black body radiation from the cooling panel, respectively. All of the radiation terms are weighted by the wavelength-dependent emissivity $\varepsilon$ (Fig. 7d) and are integrated over all of the wavelengths and the hemispheric angles. Due to the temperature dependence on the black body radiation $R_{bb}(T, \lambda) = \frac{2\pi^2 h c^2}{\lambda^5 (e^{hc/(\lambda k_B T)} - 1)}$,

where $h$ is the Plank's constant, $c$ is the speed of light and $k_B$ is the Boltzmann constant, the different spectral regions of emissivity are coupled with the spectrum of the radiation at 5800 K (i.e. $R_{sun}$) and the radiation from the surrounding temperature (i.e. $R_{atm}$, $R_m$ and $R_{bb}$) for $Q_r$. In other words, the low emissivity (or high reflectance) in visible/NIR reduces heat absorbed by $R_{sun}$ and the high emissivity in IR further facilitates heat dissipation by $R_{bb}$ for radiative cooling. For realistic simulations, the model considered the exact 3D geometries of the experimental configuration in the backyard (Supplementary Fig. 12a), incorporating $R_{sun}$ measured near the testbed (Fig. 7a) and $T_{amb}$ obtained at far away distance with elevated height from Purdue Agriculture Automated Weather Station (www.iclimate.org). Using heat transfer with a surface-to-surface module of COMSOL Multiphysics (5.2a version), we computed 3D temperature distributions by each emissive material (i.e. native silk, aluminium and carbon black paint-coated aluminium) (Fig. 7c) and temperature changes as a function of time (Fig. 7a; Supplementary Fig. 13). The current single multiphysical model analysed the interplay between the sunlight reflection/thermal emission and the radiative cooling effect, all of which are in good agreement with the experimental measurements through the day and night under a clear sky.

**Design and thermal measurements of biophotonic coolers**. In the natural environments, interrogation of radiative heat transfer is challenging, because several heat transfer mechanisms are coupled in a single system (Eq. (4)), increasing the complexity and uncertainty of thermal measurements. To selectively characterize passive radiative cooling in native silk, we minimized the measurement uncertainty by reducing the thermodynamic complexity in the following design of the experimental configuration containing a 200-mm-diameter cooling panel (Supplementary Fig. 12a). (i) The mutual radiation $R_m$ coming from other objects was minimized by exposing the cooling panel to the sky in the outdoor area. In this case, the planer geometry of the silk cooling panel is important to ensure the radiation direction within a 240-mm aperture of the testbed. For the planer surface, the 3D oval shell structure of silk cocoons was partitioned into 10 mm-diameter circular patches, which have a maximal surface curvature of 0.1 mm$^{-1}$, and the patches were continuously assembled with a thermally conductive adhesive (Omega OB-101), mimicking a cycloid pattern of fish scale (Fig. 7b). (ii) The heat load on the testbed was minimally conductive to the cooler by isolating all of the contact areas of the cooling panel from the testbed. For the isolation of contact, the cooling panel was placed on a polystyrene base and was supported inside a clear acrylic box, in which the narrow top edges were limitedly contacted to the back surface of a wooden testbed. The heat load on the testbed and the polystyrene base were also reduced by shielding with a layer of aluminized Mylar. (iii) The convective heat loss (or the convective heat transfer coefficient $h_c$) was minimized by sealing an air flow using the acrylic box and a low-density polyethylene (LDPE) film, which has high transparency in the entire visible/NIR/IR regions (Supplementary Fig. 12c). If the cooling panel was exposed to natural or strong convection (i.e. $h_c > 7.5$ W m$^{-2}$ K$^{-1}$), the overall heat transfer mechanism wasdominated by the convection and the radiative cooling effect of the silk panel at the peak sunlight was hidden beyond the sudden temperature drop by the convective heat loss (Supplementary Fig. 12b). By linear regression with the measured data, we found that LDPE sealing indeed suppressed air flow around the cooling panel, returning $h_c = 3$ W m$^{-2}$ K$^{-1}$. In turn, the effect of passive radiative cooling became prominent to be observable. When the convection was further suppressed (i.e. $h_c = 1$ W m$^{-2}$ K$^{-1}$), the silk panel can be cooled down to 30 °C theoretically below the temperature of the aluminium panel due to radiative heat exchange. For outdoor experiments, we assembled all of the components in a wood structure. A resistance temperature sensor (Omega SA1-RTD-120) was integrated at the centre immediately below the back surface of the cooling panel with the thermally conductive adhesive. To better compare with the performance of native silk's cooling, a 200-mm-diameter aluminium and carbon black paint-coated aluminium panels were placed as other reference testbeds. In a series of sunny day and night with a clear sky, we measured the steady-state temperature of these testbeds during late summer in West Lafayette, Indiana. Near the testbed, the solar irradiance $R_{sun}$ was monitored using a pyranometer (Kipp & Zonen CMP3). To be exposed with the maximum solar power, the outdoor testbeds and the pyranometer were tilted 50 °C to south, accounting for the solar elevation angle of the measurement day in October 2016.

**Data availability**. The data that support the findings of this study are available from the corresponding author upon reasonable request.

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

## Acknowledgements

We acknowledge Vladimir Shalaev, Hui Cao and Johannes de Boer for their insightful commentary; Chia-Ping Huang, Robert Seiler, Christopher Gilpin and Bradley Thiel for electron microscopy of silk fibres. This work was supported by Cooperative Research Program for Agriculture Science & Technology Development (PJ012089) from Rural Development Administration, Republic of Korea and Asian Office of Aerospace Research and Development (FA2386-16-1-4114 and FA2386-17-1-4072) from U.S. Air Force Office of Scientific Research, USA.

## Author contributions

S.H.C. and Y.L.K. mainly developed the experimental designs and analysed the data. S.H.C., M.A.V.-O., H.G., W.C. and Y.L.K. conducted the optical experiments. S.H.C., Z.K., A.M.U and Y.L.K. conducted the thermal experiments. S.-W.K., S.-R.K., K.-H.C. and T.-W.G. prepared the silk specimens and conducted the biological characterizations. Y.L.K. conceived the idea on the physical aspects and directed the research. S.H.C. and Y.L.K. mainly wrote the manuscript. All of the authors discussed the results and participated in writing the manuscript.

## Additional information

**Competing interests:** The authors declare no competing financial interests.

