## [Peer Review File(PDF 193 kb) · Nature Communications]

Reviewers' Comments:

Reviewer #3:

Remarks to the Author:

I think that the authors have gone very far in demonstrating experimentally that they indeed observe quasi 2D localised modes, where the lateral disorder plays a crucial role in trapping the light, while at the same time allowing for propagation in the third direction. While totally unexpected in biological systems, I think that the authors make a convincing case. One has to add to this that they calculate the relevant parameters and demonstrate that they are in the right range for the effect to occur.

The cooling application is elegant and could have significant biological relevance. Also here the authors make a convincing case, based on both (new) experimental results and theoretical reasoning. They seem to understand the literature on this topic very well and apply that in their studies.

Impact wise, the work has a real relevance for biology and one could expect people to find similar effects also in other biological systems in the future.

In my view the paper can be published in its present form.

Reviewer #4:

Remarks to the Author:

Second Report on "Biogenic light localization proximal to the Anderson regime reveals a self-cooling function of native silk"

I read the manuscript with interest and I am very pleased by the authors constructive response. The included application is quite interesting and I think the manuscript in this improved form is suitable for Nature Communication.

I suggest to publish the paper as it is.